# The Relationship between Serum miRNAs and Early Mortality in Multiple Myeloma Patients Treated with Bortezomib-Based Regimens

**DOI:** 10.3390/ijms24032938

**Published:** 2023-02-02

**Authors:** Anna Puła, Paweł Robak, Dariusz Jarych, Damian Mikulski, Małgorzata Misiewicz, Izabela Drozdz, Wojciech Fendler, Janusz Szemraj, Tadeusz Robak

**Affiliations:** 1Department of Hematology, Medical University of Lodz, 93-510 Lodz, Poland; 2Department of Hematooncology, Copernicus Memorial Hospital, 93-510 Lodz, Poland; 3Department of Experimental Hematology, Medical University of Lodz, 93-510 Lodz, Poland; 4Laboratory of Virology, Institute of Medical Biology, Polish Academy of Sciences, 93-232 Lodz, Poland; 5Department of Biostatistics and Translational Medicine, Medical University of Lodz, 92-215 Lodz, Poland; 6Department of Clinical Genetics, Medical University of Lodz, 92-213 Lodz, Poland; 7Department of Medical Biochemistry, Medical University of Lodz, 92-215 Lodz, Poland; 8Department of General Hematology, Copernicus Memorial Hospital, 93-510 Lodz, Poland

**Keywords:** multiple myeloma, early mortality, blood plasma, circulating miRNA, hematological malignancies, molecular biomarker, multiparametric model, prognosis

## Abstract

Multiple myeloma (MM) is a hematological malignancy characterized by the clonal proliferation of plasma cells in the bone marrow (BM) microenvironment. Despite the progress made in treatment, some MM patients still die within the first year of diagnosis. Numerous studies investigating microRNA (miRNA) expression patterns suggest they may be good prognostic markers. The primary aim of this study was to analyze the expression of selected miRNAs in the serum of MM patients who were later treated with bortezomib-based regimens, and to determine their potential to predict early mortality. The study was conducted in 70 prospectively recruited patients with newly diagnosed MM admitted to the Department of Hematology of the Copernicus Memorial Hospital, Lodz (Poland) between 2017 and 2021. Among them, 17 patients experienced death within 12 months of diagnosis. The expression of 31 selected miRNAs was determined using a miRCURY LNA miRNA Custom PCR Panel. The obtained clinical data included patient characteristics on diagnosis, treatment regimen, response to treatment, and follow-up. Differential expression analysis found two miRNAs to be significantly downregulated in the early mortality group: hsa-miR-328-3p (fold change—FC: 0.72, *p* = 0.0342) and hsa-miR-409-3p (FC: 0.49, *p* = 0.0357). Univariate and multivariate logistic regression analyses were performed to assess the early mortality rate. The final model consisted of hsa-miR-409-3p, hsa-miR-328-3p, age, and R-ISS 3. It yielded an area under the curve (AUC) of 0.863 (95%CI: 0.761–0.965) with 88.2% sensitivity and 77.5% specificity. Further external validation of our model is needed to confirm its clinical value.

## 1. Introduction

Multiple Myeloma (MM) is a malignant, heterogenous disease characterized by the clonal expansion of antibody-producing plasma cells, mostly with their origin in the bone marrow. In addition, symptomatic MM is associated with hypercalcemia, renal impairment, anemia, lytic destruction of bones (the ‘CRAB’ criteria) [1]. Multiple Myeloma is the second most common hematological malignancy, accounting for about 10% of hematological cancers; it has an annual incidence of 4.5–6 cases per 100,000 people [2], and a standardized rate of 6.1 per 100,000 inhabitants for male patients and 3.8 for female patients. It was estimated that 34,920 new cases and 12,410 deaths were attributed to MM in the US in 2021 [3]. The advent of novel agents, such as immunomodulatory drugs (IMiDs), proteasome inhibitors (PIs) and monoclonal antibodies (MoAbs), have markedly improved patient outcome [4,5]. However, MM is still an incurable disease with a median overall survival (OS) of approximately five years [6]. 

Despite significant improvements in MM treatment modalities, a subset of patients still experiences abbreviated responses and short survival [5,7]. Treatment is still burdened by high patient mortality early in the course of disease, which is often attributed to the combined effects of active disease and comorbid factors [8,9], such as chronic heart failure or diabetes mellitus. Early mortality (EM) is often defined as death within two to twelve months of diagnosis [10,11,12,13]. In some studies, cut-off values of two or six months are also used [8,9,14,15,16,17,18]. It is estimated that 4% to 25% of MM patients die within one year of diagnosis, depending on the clinical trial [19,20,21,22,23,24,25]. To maximize treatment outcomes, it is important that patients receive optimal treatment depending on the disease characteristics and patient status. Because miRNAs play important roles in such a widespread range of processes, and their deregulation can lead to pathological conditions, they became a focus of numerous studies. The role of miRNAs in both normal and cancerous conditions was extensively reviewed by Gebert et al. [26]. 

An increasing number of studies underline the role of miRNAs in the pathogenesis of MM and their potential role in its diagnosis and prognosis [27,28]. It was previously found that the expression of several micro RNAs (miRNAs), such as miR-15a and miR-16, is markedly decreased in MM patients while those of miR-21 and miR-221 are strongly increased. In addition, several miRNAs, are associated with drug resistance, and the miRNA expression patterns can be used as prognostic markers. Papanota et al. proposed a miRNA signature to facilitate MM bone disease diagnosis and provided evidence of the prognostic role of let-7b-5p and miR-335-5p as non-invasive prognostic biomarkers in MM [29]. However, no study has yet fully examined the types of miRNA present in MM patients experiencing early death. Identifying the miRNAs linked to early mortality may distinguish very-high-risk patients and improve proper treatment selection. Therefore, the present study analyzes selected serum miRNA expression patterns in the serum of newly diagnosed MM patients subsequently treated with bortezomib-based regimens.

## 2. Results

The study was performed in 70 previously untreated MM patients with a mean age at diagnosis of 65 ± 10.97 years including 34 men and 36 women. Among them, 17 patients experienced death within 12 months of diagnosis. All early deaths were MM-related (advanced disease, progression, or severe infection). Demographic, clinical and laboratory characteristics of the 70 MM patients enrolled for the study can be found in Table 1. The lack of transplant or maintenance in the early mortality group is only a result of early mortality and the fact that patients did not survive a sufficient period of time in order to receive further therapy.

Early mortality was significantly associated with older age (72.61 vs. 62.57 years, *p* = 0.0007) and male sex (*p* = 0.04). No differences in R-ISS distribution or CRAB symptoms were found between the groups. Similarly, no significant trends in percentage of myeloma infiltration in the bone marrow were noted regarding laboratory results, including LDH, serum M protein, albumin, CRP, and uric acid. 

In differential expression analysis, two miRNAs were significantly downregulated in the early mortality group—hsa-miR-328-3p (fold change—FC: 0.72, *p* = 0.048) and hsa-miR-409-3p (FC: 0.49, *p* = 0.037). A volcano plot of miRNAs with significant differences in expression is shown in Figure 1. The results of differential expression analysis for all miRNAs can be found in Appendix A.

Univariate logistic regression analyses were performed to assess the early mortality occurrence. Similarly, hsa-miR-328-3p (OR 0.486, 95%CI: 0.239–0.99, *p* = 0.0467) and hsa-miR-409-3p (OR 0.69, 95%CI: 0.487–0.976, *p* = 0.036) were acting protectively in relation to the occurrence of early mortality. The results of univariate logistic regression analyses for all miRNAs can be found in Appendix A.

A multivariate logistic regression analysis was also performed. A forward stepwise and backward stepwise selection approach were used to restrict the model to the most predictive miRNAs together with clinical features. Forward stepwise selection yielded a final model with the lowest AIC value. In the multivariate logistic regression analysis, the two miRNAs retained their significance. The final model (Table 2) consisted of hsa-miR-409-3p (OR 0.67, 95%CI: 0.43–1.05, *p* = 0.08), hsa-miR-328-3p (OR 0.39, 95%CI: 0.17–0.89, *p* = 0.03) and age (OR 1.15, 95%CI: 1.05–1.25, *p* = 0.001). The combination of hsa-miR-409-3p, hsa-miR-328-3p, and age at diagnosis could discriminate patients with high risk of early mortality.

A receiver operating characteristics (ROC) analysis for the model yielded an area under the curve (AUC) of 0.84 (95%CI: 0.73–0.95). The ROC curve is shown in Figure 2.

In the analysis of different cytogenetics risk groups, it was found that the expression of hsa-miR-627-5p was significantly different between R-ISS groups (*p* = 0.001) with its lower expression in the R-ISS I group compared to R-ISS II (*p* = 0.015) and R-ISS III group (*p* = 0.001). A box plot of the hsa-miR-627-5p expression in different R-ISS groups can be found in Figure 3A. Similarly, the expression of hsa-miR-627-5p was significantly higher in the high-risk cytogenetics group compared to standard-risk group (*p* = 0.004, Figure 3B).

Normalized ΔCt for all samples with class assignments and crucial clinical data are provided as Appendix A.

## 3. Discussion

In the present study, we analyzed the expression of selected miRNAs in the serum of MM patients who were later treated with bortezomib-based regimens, and to determine their potential to predict early mortality. Despite spectacular advancements in therapy, early mortality remains a significant problem in MM patient care. Clinical trials have found between 4% and 25% of MM patients die within one year of diagnosis [19,20,21,22,23,24,25]. However, data on early mortality (EM) outside the context of clinical trials remain extremely scarce. The mortality rate during the first year after diagnosis has reduced considerably over the years [13]. The most common causes of early death are reportedly infections, progression of MM, cardiovascular diseases, and renal failure [9,30]. The phenomenon of EM is often attributed to combined effects of active disease and comorbid factors [31]. However, various clinical features such as beta 2-microglobulin, serum lactate dehydrogenase level, performance status, age at diagnosis, ISS, history of cardiovascular diseases or diabetes mellitus, presence of bone disease, and diminished renal function have been associated with EM by various studies [8,30,32]. 

To date only two approaches have been accepted in the prognosis of MM: the International Staging System (ISS), which combines albumin and β2-microglobulin levels, and the presence of chromosomal abnormalities detected via FISH (fluorescence in situ hybridization) [6,33]. However, it is not clear if the prognostic factors identified in the era of older drugs are still of value in the current era of novel therapies. One new approach for predicting clinical outcome is circulating serum miRNA expression. Our findings from the presented study demonstrate that a combination of certain miRNA levels together with selected clinical factors may predict the occurrence of early mortality in newly diagnosed MM patients. This is the first study to evaluate the potential of miRNA expression as a predictor of early death in this population. 

Manier et al. found that let-7b and miR-18a were significantly associated with both progression-free survival (PFS) and overall survival (OS) in univariate analysis and remained statistically significant after adjusting for the ISS and adverse cytogenetics in multivariate analysis [34]. Hao et al. reported the downregulation of miR-19a to be associated with significantly decreased PFS and OS [35]. Furthermore, miR-125b-5p may be another potential clinical biomarker for MM associated with significantly shorter event-free survival (EFS) in myeloma patients [36]. In a systematic review and meta-analysis examining the prognostic value of miRNAs in patients with MM, Xu et al. reported that the upregulation of miR-92a and downregulation of miR-16, miR-25, miR-744, miR-15a, let-7e, and miR-19b expression were associated with poor prognosis [37].

Following on from our previous study examining the value of serum microRNA expression signature in predicting refractoriness to bortezomib-based therapy in multiple myeloma patients [38], the present work attempted to determine whether selected microRNAs can have predictive value regarding early mortality. We identified two miRNAs, hsa-miR-328-3p and hsa-miR-409-3p, to be significantly downregulated in the early mortality of MM patients. Both hsa-miR-409-3p and hsa-miR-328-3p have already been investigated in various cancers. The dysregulation of miR-409 has been detected in many neoplasms, including gastric cancer, prostate cancer, bladder cancer and lung adenocarcinoma [39,40,41,42,43]; miR-409-3p reportedly regulates cell proliferation and invasion by targeting zinc-finger E-box-binding homeobox 1 (ZEB1). Overexpression of miR-409-3p inhibits cellular proliferation and was found to suppress cellular migration and invasion in vitro and in vivo in breast cancer and osteosarcoma [44,45]. It may also have a potential tumor suppressor function in cervical malignancies by regulating the HPV16/18-E6 mRNA levels [46]. Downregulation of microRNA-409-3p promotes aggressiveness and metastasis in colorectal cancer, while its overexpression sensitizes cells to oxaliplatin by inhibiting Beclin-1-mediated autophagy [47,48].

It was reported that miRNA-328 may decrease chemoresistance in glioblastoma cancer cells and breast cancer cells by down-regulating the ABCG2 gene [49,50]. Downregulation of miR-328-3p was observed in colorectal cancer, while its overexpression reversed the process of drug resistance and inhibited cell invasion [51]. In contrast, studies investigating the role of miR-328-3p in head and neck squamous cell carcinoma and in ovarian cancer found its overexpression to be associated with more invasive disease [52,53].

Although the risk of MM increases with age, no data could be found about the association between hsa-miR-409-3p or hsa-miR-328-3p expression and age or senescence in existing literature. The role of these miRNAs in the pathogenesis of multiple myeloma to date also remains unknown. However, hsa-miR-627-5p has been investigated in other cancers including hepatocellular carcinoma and colorectal neoplasms [54,55]. Our findings demonstrate that serum expression of hsa-miR-409-3p and hsa-miR-328-3p is downregulated in patients experiencing early mortality. These results were used to generate a multiple regression model that may have the potential to predict EM. By extending known prognostic systems with more comprehensive molecular data, such as miRNA expression, it may be possible to increase the chance of identifying high-risk patients. Additionally, our analysis of miRNA expression regarding R-ISS and cytogenetic groups revealed a different marker than that used in early mortality, namely hsa-miR-627.5p. This observation suggests that early mortality and cytogenetic risk may be characterized by different molecular marker profiles. However, the number of our patients was relatively small, considering the whole MM population. Further external validation of our model is necessary to confirm its clinical value.

## 4. Materials and Methods

### 4.1. Patients and Treatment

The study was performed in previously untreated MM patients who were qualified for treatment with bortezomib-based regimens based on the decision of the medical board. Sets of clinical and laboratory data were obtained: patient characteristics on diagnosis, treatment regimen, response to treatment and follow-up. For inclusion in the study, the patients needed to have newly diagnosed, untreated multiple myeloma, and measurable disease confirmed by the presence of monoclonal protein in blood or Bence–Jones proteinuria. Patients were treated at the Department of Hematology, Copernicus Memorial Hospital, Lodz, Poland in the years 2017–2021. Responses to treatment, progression free survival (PFS), and overall survival (OS), were assessed according the International Myeloma Working Group (IMWG) criteria [56,57]. Early mortality was defined as death within one year of diagnosis. The study and experimental protocol were conducted according to good clinical and laboratory practice rules and the principles of the Declaration of Helsinki. All procedures were approved by the local ethical committee (The Ethical Committee of the Medical University of Lodz, No RNN/103/16/KE). All patients included in the study gave written informed consent for all examinations and procedures.

### 4.2. Isolation of miRNA

Venous blood samples were collected from previously untreated MM patients in serum separating tubes. The samples were processed within two hours of collection by centrifugation at 2400× *g* for 10 min, as described previously [38]. Serum samples were stored at −80 °C. Isolation of cell-free total RNA, including miRNA, was performed from serum using the miRNeasy Serum/Plasma Advanced Kit (Qiagen, Hilden, Germany) according to the manufacturer’s protocol. Briefly, 200 µL of serum was mixed with Buffer RPL to ensure a complete lysis and to release and stabilize RNA from plasma proteins and extracellular vesicles. To allow for normalization of sample-to-sample variation in RNA isolation and to control the quality of RNA isolation, before purification, each serum sample was spiked with 22 nt synthetic miRNAs added to the RPL Buffer, using the RNA Spike-In Kit, For RT (Qiagen, Hilden, Germany), according to the manufacturer’s protocol. Briefly, each sample was spiked with UniSp2 (2 fmol), UniSp4 (0.02 fmol), UniSp5 (0.00002 fmol), each at 100-fold reductions in concentration. The sample was then mixed with Buffer RPP to precipitate proteins and inhibitors and then centrifuged. Isopropanol was added to the supernatant to provide the appropriate conditions for RNA molecules (>18 nucleotides) to bind to the silica membrane. The sample was then applied to a RNeasy UCP MinElute spin column, where RNA, including miRNA, binds to the membrane and other contaminants are washed away in subsequent wash steps. In the final step, total RNA (>18 nucleotides) was eluted using 20 µL of RNase-free water. The RNA quality was determined with Agilent High Sensitivity RNA ScreenTape using 2200 TapeStation (Agilent Technologies, Santa Clara, CA, USA). Only the samples with RIN > 7.0 were further processed. Directly after the isolation, RNA was subjected to reverse transcription.

### 4.3. Reverse Transcriptase Reaction 

The cDNA was synthesized from the obtained total RNA including mature miRNAs (<200 bp), using the miRCURY LNA Reverse Transcription Kit (Qiagen, Hilden, Germany), according to the guidelines provided by the manufacturer. Mature miRNAs were polyadenylated by poly(A) polymerase and reverse transcribed into cDNA using oligo-dT primers. Polyadenylation and reverse transcription were performed in parallel in the same tube. The oligo-dT primers have a 30-degenerate anchor that allows the amplification of mature miRNA in the real-time PCR step. The total volume of 2 µL of the isolate was added to the reaction tube containing 2 µL of 5× miRCURY RT Reaction Buffer, 1 µL of 10× miRCURY RT Enzyme Mix and filled with RNase-free water to make up a final volume of 10 µL reaction mix. Two synthetic miRNAs: UniSp6 (0.075 fmol) and cel-miR-39-3p (0.001 fmol) were used as a positive control for cDNA synthesis. The reaction took place at 42 °C for 60 min, followed by inactivation at 95 °C for 5 min using a TProfessional thermal cycler (Biometra, Analytik Jena, Jena, Germany). The cDNA was stored at −20 °C until further use.

### 4.4. Determination of microRNA Expression 

Before the miRNA expression analysis, the RNA samples were subjected to quality control using the miRCURY LNA miRNA QC PCR Panel (Qiagen, Hilden, Germany), according to the manufacturer’s protocol. The Panel contains 12 predefined assays for miRNAs that are expressed in a wide range of sample types used for evaluating samples in a biological context. Two of the miRNAs in the panel (miR-451 and miR-23a) were used to evaluate the level of hemolysis in serum samples. If the ΔCq (miR-23a—miR-451) is close to or higher than 7, there was an increased risk that the samples are affected by hemolysis, and these samples were rejected from the further analysis (in case of high levels of hemolysis, miRNAs from red blood cells will make a significant contribution to the overall miRNA expression) [58]. Only the samples with ΔCq(miR-23a-miR-451) < 7.0 were further processed.

The expression of miRNAs was determined in 70 patients using miRCURY LNA miRNA Custom PCR Panel (Qiagen, Hilden, Germany). The selection of miRNAs included in this study was performed based on our previous analysis of 752 miRNAs [38] and was based on the *p* value significance from the univariate logistic regression analysis for overall survival (unpublished results from [38]). The miRNA expression was performed for 31 miRNAs in duplicate. The list of selected miRNAs together with their GeneGlobe ID and mature miRNA sequence can be found in Appendix A.

### 4.5. Real-Time PCR (qPCR) 

A premix of 3 µL of cDNA template (diluted 1:30), 5 µL of 2X miRCURY LNA SYBR Green PCR Master Mix and filled with RNase-free water to the final volume of 10 µL, was aliquoted into the PCR plate. Real-time PCR was performed on a LightCycler480 II Real-Time PCR System (Roche, Pleasanton, CA, USA). The reaction was performed at 95 °C for 2 min, followed by 55 amplification cycles at 95 °C for 10 s and 56 °C for 1 min. Fluorescence signal (the excitation and emission. Maxima of SYBR^®^ Green I: 494 nm and 521 nm, respectively) was measured after each cycle. Melting curve analysis was performed after each qPCR to assess the specificity of the amplification. Absolute quantification of miRNA was determined using the LightCycler^®^ 480 Software, Version 1.5 (Roche, Mannheim, Germany). Relative quantification of miRNA was determined by comparative Ct method.

### 4.6. Statistical Analysis

Data normalization was performed based on the mean expression value of two miRNAs in a given sample (miR-23b-3p and miR-152-3p). These miRNAs, according to the NormiRazor tool, proved to be the second most stable normalization pair in the currently analyzed group of patients and was chosen on the basis of our previous study where a triplet of miR-23b-3p, miR-151a-5p and miR-152-3p was chosen for the whole panel [38]. However, after a thorough analysis, in the currently presented study we chose to use only two miRNAs instead of three for the normalization. The following formula was used to calculate the normalized Ct values: Normalized ΔCt = mean Ct of miR-23b-3p, and miR-152-3p—Ct miRNA. A table and a plot presenting stability indexes for the top 10 normalizing pair can be found in Appendix A.

The study group and intragroup association were described based on assumption testing using the Chi-squared or Fisher’s exact test. The normality of the distribution of continuous variables was verified with the Shapiro–Wilk test. For continuous variables, the difference between two groups was evaluated using a two-sided independent Student’s *t*-test when the data was normally distributed, and with the Wilcoxon–Mann–Whitney test when it was not, or if the variable was ordinal. These statistical tests were also used for the differential expression analysis comparing the normalized expression values in two experimental groups.

For a more comprehensive analysis, a logistic regression model was generated. The early mortality was used as outcome and the clinical features (demographics, blood count, biochemical tests, immunofixation, electrophoresis, cytogenetic results, treatment) together with miRNAs expression as predictors. The Student’s *t*-test was used to preselect variables for the development of the classification model. The miRNAs significant in the DE analysis were chosen as candidate variables in this step. Both a univariate model for each of the selected miRNA and a multivariate model that included all selected predictors were estimated. Forward stepwise and backward stepwise selection approaches were used to restrict the model. The lowest AIC (The Akaike information criterion) value was chosen as the determinant of the best final model. Receiver operating characteristics (ROC) and area under the curve (AUC) analysis were performed to determine the predictive power of the model and its ability to accurately predict early mortality.

The Kruskal–Wallis test with Bonferroni post hoc Wilcoxon–Mann–Whitney tests and the one-way ANOVA test with Bonferroni post hoc *t*-tests were used to compare the miRNA expression between different R-ISS groups depending on normality and homogeneity of variance. The Wilcoxon–Mann–Whitney test and the Student’s *t*-test were used to compare the miRNA expression between high-risk cytogenetics and standard-risk patients [1,6,7]. 

All statistical analyses were conducted using Statistica Version 13.1 (TIBCO, Palo Alto, CA, USA) and R programming language (version 4.1.3) software. The OmicSelector R package was partly used in the analysis [59] *p* values lower than 0.05 were considered statistically significant. As all the analyses were preplanned, no correction for multiple comparisons was applied.

## 5. Conclusions

In conclusion, we identified two miRNAs, hsa-miR-409-3p and hsa-miR-328-3p, that are independent factors related to early mortality in MM patients. We used our results to generate a multiple regression model that may have the potential to predict early mortality. However, further study is needed to confirm our findings. 

## Figures and Tables

**Figure 1 ijms-24-02938-f001:**
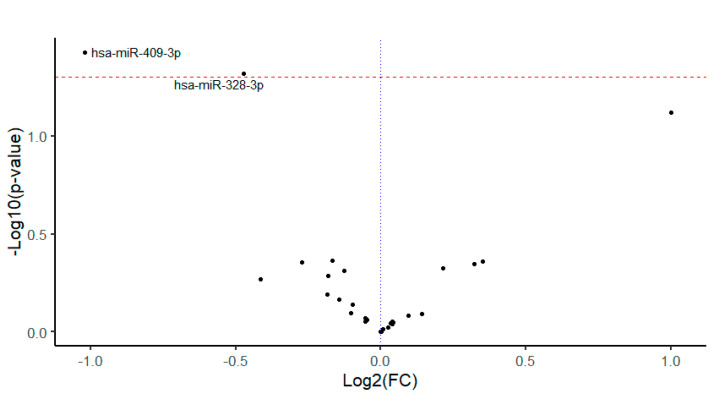
Volcano plot of 29 investigated miRNAs. Significant miRNAs have been labeled.

**Figure 2 ijms-24-02938-f002:**
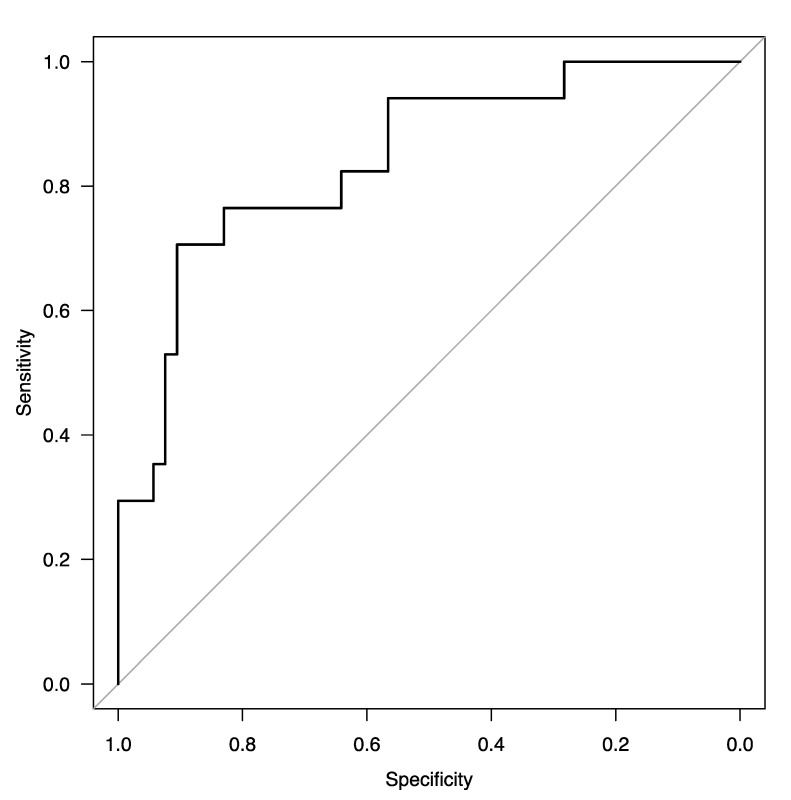
ROC curve of the final model for assessing factors associated with early mortality in MM patients.

**Figure 3 ijms-24-02938-f003:**
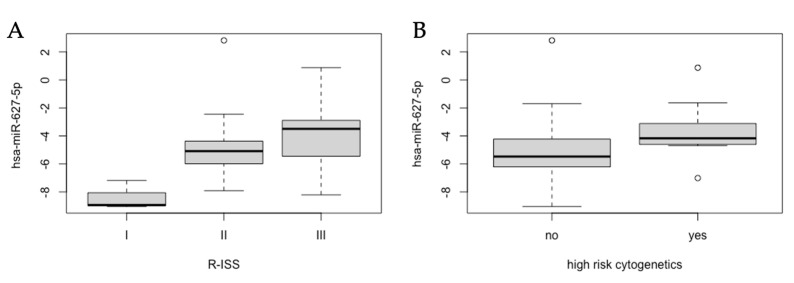
A box plot of the hsa-miR-627.5p expression in different R-ISS groups (**A**) and in different cytogenetics groups ((**B**)—high-risk cytogenetics vs. standard-risk cytogenetics).

**Table 1 ijms-24-02938-t001:** Patient characteristics. LCD—light chain disease; R-ISS—Revised International Staging System. ^#^—Pearson’s Chi-squared test, *—Fisher’s Exact Test.

Characteristic (n = 70)		No. (%)			
		All	Early Mortality	Control	*p*-Value
**No. of Patients**		70	17	53	
**Age, Years, Mean (SD)**		65 (10.97)	72.61 (7.81)	62.57 (10.77)	0.0007
**sex**	Male, n (%) Female, n (%)	34 (48.57) 36 (51.43)	12 (70.59) 5 (29.41)	22 (41.51) 31 (58.49)	0.04 ^#^
**R-ISS**	Stage I, n (%) Stage II, n (%) Stage III, n (%)	3/57 (5.26) 34/57 (59.65) 20/57 (35.09)	1/17 (5.88) 9/17 (52.94) 7/17 (41.18)	2/57 (3.51) 25/57 (43.86) 13/57 (22.81)	0.8 *
**type**	IgA kappa, n (%) IgA lambda, n (%) IgG kappa, n (%) IgG lambda, n (%) LCD kappa, n (%) LCD lambda, n (%)	10 (14.29) 5 (7.14)29 (41.43) 13 (18.57) 6 (8.57) 7 (10)	2 (11.76) 0 (0) 7 (41.18) 5 (29.41) 2 (11.76) 1 (5.88)	8 (15.09) 5 (9.43) 22 (41.51) 8 (15.09) 4 (7.55) 6 (11.32)	0.68 *
**cytogenetics**	del17p, n (%) t(4;14), n (%) t(14;16), n (%) t(14;20), n (%) t(11;14), n (%) del1p, n (%) amp1q, n (%) del13q, n (%)	6/59 (10.17) 3/58 (5.17) 2/58 (3.45) 0/55 (0) 5/54 (9.26) 5/56 (8.93) 25/57 (43.86) 8/30 (26.67)	2/17 (11.76) 0/17 (0) 0/17 (0) 0/16 (0) 3/16 (18.75) 2/16 (12.5) 7/16 (43.75) 3/11 (27.27)	4/42 (9.52) 3/41 (7.32) 2/41 (4.88) 0/39 (0) 2/38 (5.26) 3/40 (7.5) 18/41 (43.9) 5/19 (26.32)	1.0 *0.55 * 1.0 *- 0.15 * 0.62 * 0.99 ^#^ 1.0 *
**anemia at diagnosis, n (%)**		31 (44.29)	8 (47.06)	23 (43.4)	0.79 ^#^
**bone disease, n (%)**		51/67 (76.12)	13/16 (81.25)	38/51 (74.51)	0.74 *
**creatinine > 2 mg/dL at diagnosis, n (%)**		13 (18.57)	4 (23.53)	9 (16.98)	0.72 *
**hypercalcemia at diagnosis, n (%)**		24 (34.29)	7 (41.18)	17 (32.08)	0.49 ^#^
**treatment, n (%)**	VCD VTD VMP VD VRD IsaVRD DaraVRD autoSCT	48 (69.57) 5 (7.14) 8 (11.43) 6 (8.57) 1 (1.43) 1 (1.43) 1 (1.43) 27 (38.57)	9 (52.94) 0 (0) 4 (23.53) 3 (17.65) 0 (0) 1 (5.88) 0 (0) 0 (0)	39 (73.58) 5 (9.43) 4 (7.55) 3 (5.66) 1 (1.89) 0 (0) 1 (1.89) 27 (50.94)	0.09 ^#^ 0.32 * 0.09 * 0.15 * 1.0 *0.25 * 1.0 * <0.001 *
**maintenance**		13/65 (20)	0/65 (0)	13/65 (20)	0.03 *

**Table 2 ijms-24-02938-t002:** Final multivariate regression model for predicting early mortality in MM patients.

Variable	OR	95%CI		*p*
		Lower	Upper	
hsa-miR-409-3p	0.67	0.43	1.05	0.08
hsa-miR-328-3p	0.39	0.17	0.89	0.03
Age	1.15	1.05	1.25	0.001

## Data Availability

The data presented in this study are available from the corresponding author on request.

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
