# Peer review of "The Relationship between Serum miRNAs and Early Mortality in Multiple Myeloma Patients Treated with Bortezomib-Based Regimens"

_ijms, 2023, doi:10.3390/ijms24032938_

Round 1
Reviewer 1 Report
In this manuscript, the authors explore the prognostic importance of miRNAs on early mortality after MM diagnosis and show that under expression of hsa-miR-409-3p and hsa-miR-328-3p in combination with other clinical markers has a high prognostic impact on adverse outcome. While the topic is of interest and the design as well as technical methods seem sound, there is some concern with the analysis and interpretation of the data. Addressing the following concerns would be helpful in improving the quality of the paper.
-The authors claim that early deaths were MM-related (line 216). Does that mean that patient had progression of disease? Is the under expression of the particular two miRNAs also associated with PFS?
- Based on Table 1 it appears that patients with early mortality were clearly under-treated, as none of them has received a transplant nor maintenance. That seems to be a possible confounder in the approach and analysis and a reason why patient's MM did poorly. Was the treatment approach (transplant and/or maintenance) included in the multivariate analysis?
-It is not evident which factors were included in the multivariate analysis and that should be mentioned in the manuscript.
- The model in Figure 2 is not well explained and it is not obvious why the authors chose R-ISS (which was not significant in the multivariate analysis)and age in combination with the two miRNAs. Please elaborate.
-In Table 1, cytogenetics, could the authors please confirm if it is dup1q or amp1q? 40% seems a very high number of amp1 for newly diagnosed MM.
Author Response
Reviewer 1
In this manuscript, the authors explore the prognostic importance of miRNAs on early mortality after MM diagnosis and show that under expression of hsa-miR-409-3p and hsa-miR-328-3p in combination with other clinical markers has a high prognostic impact on adverse outcome. While the topic is of interest and the design as well as technical methods seem sound, there is some concern with the analysis and interpretation of the data. Addressing the following concerns would be helpful in improving the quality of the paper.
-The authors claim that early deaths were MM-related (line 216). Does that mean that patient had progression of disease? Is the under expression of the particular two miRNAs also associated with PFS?
Response: Thank you for your comment. The expression “MM-related” is now explained in the text. The expression of the two miRNAs was not associated with PFS.
- Based on Table 1 it appears that patients with early mortality were clearly under-treated, as none of them has received a transplant nor maintenance. That seems to be a possible confounder in the approach and analysis and a reason why patient's MM did poorly. Was the treatment approach (transplant and/or maintenance) included in the multivariate analysis?
Response: Thank you for your question. The lack of transplant or maintenance is only a result of early mortality and the fact that patients did not survive a sufficient period of time in order to receive further therapy. The treatment approach was not different, only the survival.
-It is not evident which factors were included in the multivariate analysis and that should be mentioned in the manuscript.
Response: Thank you for your comment. The factors included in the multivariate analysis are now mentioned in the manuscript in the methods section.
- The model in Figure 2 is not well explained and it is not obvious why the authors chose R-ISS (which was not significant in the multivariate analysis) and age in combination with the two miRNAs. Please elaborate.
Response: An error occurred in one of the sentences in the text. We are sorry for the mistake. It is now corrected. The intention of the model was to separate patients into two groups by the occurrence of early mortality. The final model consists of two miRNAs and age.
-In Table 1, cytogenetics, could the authors please confirm if it is dup1q or amp1q? 40% seems a very high number of amp1 for newly diagnosed MM.
Response: Thank you for your comment. We once again checked all the cytogenetics results and we confirm that in our analyzed population the frequency of amp1q is high (over 40%).
Reviewer 2 Report
Dear authors, it was my pleasure to review your recent work regarding miRNA in MM.
The results and reasoning is clearly presented and leads to further insights regarding biomarkers in a complicated disease. If the predictive model could be validated the work could also lead to an impact on the clinical practice.
I have minor comments only:
1. It is stated that 31 preselected miRNAs are studied and that they are based on your previous work (ref 32). However, the study cited finds 21 significant miRNAs. Can you please elaborate on the selection process for the miRNAs? (section 2.4) In sup table one 28 miRNAs are listed and 3 are used for reference and normalization purposes. Perhaps it is more accurate to list it as 28 analysed miRNAs that are used to answer questions regarding the patient population.
2. In the intro it is listed as 69 patients, in sup table 5 it is 70. Please correct.
3. 13 patients lack R-ISS data, how were they handled in the multivariate regression model (table 2). Please elaborate on the possible impact in relation to how they were handled.
4. In the discussion several other miRNAs that have had significant correlation with the patient population previously are listed (e.g., MIR-18a, 19a, 125b-5p etc.). Why were they not included in the current analysis (prior knowledge etc)? A problem with research of this type is often that finding are not validated and you highlight the need for this approach for your own findings as well. It would be interesting, when you have gathered a nice prospective material to test some of the previously described miRNAs in the literature.
5. In relation to question 1, it is stated that adjustment for multiplicity testing is not needed due to planned experiments. Please expand this reasoning. E.g., is not needed because the pre-selection testing was carried out with rigorous adjustment? Other reasons? Most experiments can probably be described as planned.
Apart from the above no further comments.
Author Response
Reviewer 2
Dear authors, it was my pleasure to review your recent work regarding miRNA in MM.
The results and reasoning is clearly presented and leads to further insights regarding biomarkers in a complicated disease. If the predictive model could be validated the work could also lead to an impact on the clinical practice.
I have minor comments only:
1.It is stated that 31 preselected miRNAs are studied and that they are based on your previous work (ref 32). However, the study cited finds 21 significant miRNAs. Can you please elaborate on the selection process for the miRNAs? (section 2.4) In sup table one 28 miRNAs are listed and 3 are used for reference and normalization purposes. Perhaps it is more accurate to list it as 28 analysed miRNAs that are used to answer questions regarding the patient population.
Response: Thank you for your question. As stated in the manuscript the selection was based on our previous work. However, the analysis of univariate logistic model in the context of overall survival irrespective of bortezomib resistance was not the subject of our previous paper. The selection was performed based on some unpublished data. It is now stated in the manuscript. The normalization was performed on the basis of two miRNA. The mistake in supplementary table 1 was corrected.
2.In the intro it is listed as 69 patients, in sup table 5 it is 70. Please correct.
Response: Corrected as indicated.
3.13 patients lack R-ISS data, how were they handled in the multivariate regression model (table 2). Please elaborate on the possible impact in relation to how they were handled
Response: The final model consists of two miRNAs and age. Since the study was mostly performed in R software the multivariate stepwise selection takes only complete cases into account. The model was also checked in Statistica programme with almost the same test results. We do not think the percentage of missing values in other variables had any impact on the final model. It was counted for all cases with no missing values in the two miRNAs or age.
4.In the discussion several other miRNAs that have had significant correlation with the patient population previously are listed (e.g., MIR-18a, 19a, 125b-5p etc.). Why were they not included in the current analysis (prior knowledge etc)? A problem with research of this type is often that finding are not validated and you highlight the need for this approach for your own findings as well. It would be interesting, when you have gathered a nice prospective material to test some of the previously described miRNAs in the literature.
Response: Thank you for your question. The selection was based on the results from our previous work. We mentioned these miRNAs in the discussion section in order to draw some background. The is still no data on their association with the occurrence of early mortality.
5.In relation to question 1, it is stated that adjustment for multiplicity testing is not needed due to planned experiments. Please expand this reasoning. E.g., is not needed because the pre-selection testing was carried out with rigorous adjustment? Other reasons? Most experiments can probably be described as planned.
Response: Thank you for your valuable question. Such correction is often needed in the context of genetic data where the number of results is significantly higher than in our study. Therefore, we decided that such comment would be needed. However, this statistical problem does not concern the final model creation.
Apart from the above no further comments.

Round 2
Reviewer 1 Report
Thank you for your comments and revisions. All concerns were addressed. Please only add the explanation of why there is significant difference in transplant/maintenance between these two groups into the text (eg "the lack of transplant or maintenance is only a result of early mortality and the fact that patients did not survive a sufficient period of time in order to receive further therapy. )
Author Response
Thank you for your comments and revisions. All concerns were addressed. Please only add the explanation of why there is significant difference in transplant/maintenance between these two groups into the text (eg "the lack of transplant or maintenance is only a result of early mortality and the fact that patients did not survive a sufficient period of time in order to receive further therapy. )
Response: Thank you very much for your suggestion. We have added the sentence to the text: The lack of transplant or maintenance in the early mortality group is only a result of early mortality and the fact that patients did not survive a sufficient period of time in order to receive further therapy.
